Change of intestinal microbiota in mice model of bronchopulmonary dysplasia

Fan Tianqun 1
Lu Ling 1
Jin Rong 1
Sui Aihua 2
Guan Renzheng 1
Cui Fengjing 1
Qu Zhenghai 1 quzhenghai@163.com
Liu Dongyun 1 liudongyun007@163.com
1 Department of Pediatrics, Affiliated Hospital of Qingdao University , Qingdao , China
2 Medical Research Center, Affiliated Hospital of Qingdao University , Qingdao , China
Collado Maria Carmen
Electronic publication date: 2022 Apr 20
Publication date: 2022
Volume: 10
Electronic Location ID: e13295
Received 2021 Jul 16; Accepted 2022 Mar 28
Copyright: © 2022 Fan et al.
Copyright year: 2022
Copyright holder: Fan et al.
License: This is an open access article distributed under the terms of the Creative Commons Attribution License, which permits unrestricted use, distribution, reproduction and adaptation in any medium and for any purpose provided that it is properly attributed. For attribution, the original author(s), title, publication source (PeerJ) and either DOI or URL of the article must be cited.
License URL: https://creativecommons.org/licenses/by/4.0/

Keywords: Hyperoxia, Bronchopulmonary dysplasia, Gut microbiota, 16S rRNA

Funding: The authors received no funding for this work.

==============================
Background

Gut microbiota has been proposed to be related to the pathogenesis of pulmonary diseases such as asthma and lung cancer, according to the gut-lung axis. However, little is known about the roles of gut microbiota in the pathogenesis of bronchopulmonary dysplasia (BPD). This study was designed to investigate the changes of gut microbiota in neonatal mice with BPD.

Methods

BPD model was induced through exposure to high concentration of oxygen. Hematoxylin and eosin (H&E) staining was utilized to determine the modeling efficiency. Stool samples were collected from the distal colon for the sequencing of V3–V4 regions of 16S rRNA, in order to analyze the gut microbiota diversity.

Results

Alpha diversity indicated that there were no statistical differences in the richness of gut microbiota between BPD model group and control group on day 7, 14 and 21. Beta diversity analysis showed that there were statistical differences in the gut microbiota on day 14 (R = 0.368, p = 0.021). Linear discriminant analysis effect size (LEfSe) showed that there were 22 markers with statistical differences on day 14 (p < 0.05), while those on day 7 and 21 were 3 and 4, respectively. Functional prediction analysis showed that the top three metabolic pathways were signal transduction (PFDR = 0.037), glycan biosynthesis and metabolism (PFDR = 0.032), and metabolism of terpenoids and polyketides (PFDR = 0.049).

Conclusions

BPD mice showed disorder of gut microbiota, which may involve specific metabolic pathways in the early stage. With the progression of neonatal maturity, the differences of the gut microbiota between the two groups would gradually disappear.

Background

Bronchopulmonary dysplasia (BPD) is a chronic pulmonary disease that is commonly reported in neonates after long-term oxygen inhalation or mechanical ventilation (Jensen et al., 2019). Some children with BPD may present persistent lung function deterioration until reaching adulthood (Cassady, Lasso-Pirot & Deepak, 2020; Principi, Di Pietro & Esposito, 2018). The pathogenesis of BPD is multifactorial. To date, there are still no ideal treatment options for treating children with BPD. It has been shown that alterations in the lung microbiota are associated with several diseases, such as asthma, chronic obstructive pulmonary disease, and cystic fibrosis (Barcik et al., 2020; Cuthbertson et al., 2020; Sze, Hogg & Sin, 2014). In hyperoxia environment, the changes of mouse lung microbiota preceded the emergence of lung tissue injury (Ashley et al., 2020). A sterile mouse model shows that mice lacking the respiratory microbiota have less lung damage induced by a hyperoxia environment (Dolma et al., 2020). With the proposal of the gut-lung axis concept (Budden et al., 2017), the relationship between alterations in intestinal microbiota and BPD warrants further exploration. Bacteria, as the most abundant and important component of intestinal microbes (Arumugam et al., 2011), play a decisive role in the gut-lung axis. Therefore, we mainly investigated the alteration of gut bacteria in relation to BPD. In addition, we established a hyperoxia BPD mouse model and explored the alteration of gut bacteria in this model.

Materials and Methods

Animals

Mice, like premature infants, are in the saccular stage of lung development at birth, which can simulate the development process of lung tissue in children with BPD. Moreover, mice are very suitable to be selected as animal models because of their high gene homology with human genes, short pregnancy cycle and low test cost (Giusto et al., 2021). Pregnant mice of specific-pathogen free (SPF) Kunming mice were purchased from Sipeifu Biotech (Beijing, China; Approval No.: SCXK-2019-0010), and housed in SPF-grade environments. Experiments were implemented in the Experimental Animal Center of Qingdao University and the Medical Research Center of Affiliated Hospital of Qingdao University. The study protocols were approved by the Ethical Committee of Affiliated Hospital of Qingdao University (Approval No.: QYFYWZLL26150).

Induction of BPD and grouping

Neonatal mice were randomly divided into control group (n = 5 L) and BPD group (n = 5 L), and there are 5–6 mice in each litter. There is no consensus on the optimal oxygen for inducing BPD models (Giusto et al., 2021), and we referred to one of the development model of mice (Nardiello et al., 2017). Neonatal mice in BPD group were subject to high concentration of oxygen (80 ± 5%) for 3 weeks. In addition, the carbon dioxide generated by the mice was removed using the absorption agent. In control group, the animals were exposed to oxygen at a concentration of 21%. All the animals in both groups were kept under a temperature of 20 ± 2 °C, in a humidity of 55 ± 5%, in a light cycle of 12 h/12 h. They were all free access to breast-feeding. The mice for breeding these animals were exchanged per day to prevent oxygen toxicity.

Experimental procedure

On day 7, 14 and 21 after birth, one mouse was randomly selected from each litter and was sacrificed using cervical dislocation. Then the pulmonary tissues and stool samples were obtained. Pulmonary samples were washed with sterile PBS solution, fixed in 4% paraformaldehyde, and embedded in paraffin, followed by Hematoxylin and Eosin (H&E) staining to determine the presence of pulmonary alveolar fusion, inflammatory infiltration, pulmonary septum thickening, pulmonary tissue disorder, in order to validate the modeling.

To prevent the environmental pollution, the stool samples were obtained under sterilized conditions from the distal colon, followed by storing at −80 °C for analysis. Stool samples were subject to sequencing of V3–V4 regions of 16s rRNA (Claesson et al., 2010) based on Illumina HiSeq platform provided by BMKCloud (Beijing, China), in order to obtain the raw sequencing data of the gut microbiota. The primers were 338F (5′- ACTCCTACGGGAGGCAGCA-3′) and 806R (5′-GGACTACHVGGGTWTCTAAT-3′). Our raw data files have been uploaded online on NCBI (PRJNA743668). After that, raw reads were filtered by Trimmomatic (v0.33), and the primers were removed by Cutadapt (v1.9.1), to obtain high-quality reads. After high-quality reads were spliced by FLASH (v1.2.7), the clean reads can be obtained. Finally, clean reads can obtain effective sequences by removing chimeras through UCHIME (v4.2). The statistics of sequencing data processing process were shown in the Table S1.

We used Usearch to cluster the reads at 97.0% similarity level and obtain the operational taxonomic units (OTUs). Taking Silva as the reference database, we used naive Bayesian classifier to classify the OTUs, in order to obtain the composition of gut microbiota at various levels.

In this section, we analyzed the gut microbiota in both groups at different stages. Ace and Chao1 index were obtained under a similarity of 97%. Then species accumulation curves and Shannon–Wiener curves were established. These data could reflect the sample size and abundance between BPD model group and control group. Afterwards, PCoA data from each group were obtained, in order to validate the significance of the sample similarity. Then the biomarkers at different stages were obtained, in order to further analyze the significances at the phylum level. Finally, we tried to identify different metabolic pathways between two groups.

Statistical analysis

Data were presented as mean ± standard deviation. Alpha and Beta analyses were conducted by QIIME2 analysis platform. We conducted the LEfSe data analysis between two groups with toolkit of Python language, and the analysis of similarities (Anosim) with the vegan toolkit of R language. The metabolic pathway analysis was conducted using Picrust2 software. On the Student’s t-test, 95% confidence intervals were used. A p value of less than 0.05 was considered to be statistically significant. PFDR was p-value using false discovery rate (Noble, 2009). The analyses were drawn using GraphPad Prism 9. It can be seen from the statistical table of each classification level that 78% of bacteria at the species level are uncultured bacteria (Table 1). Therefore, we did not discuss the classification of gut microbiota at the species level, and we mostly discussed it at the phylum level.

Table 1 The statistics of uncultured bacterium.

Level	Phylum	Class	Order	Family	Genus	Species	
Count	0	610	9,442	54,658	387,267	1,552,474	
Ratio (%)	0	0.03	0.48	2.76	19.59	78.53	
Note:

Count: the effective reads of uncultured bacterium; Ratio: the percentage of count in all effective reads (n = 1,976,896).

Results

Morphological changes of lung in different groups

H&E staining indicated that the morphology of pulmonary alveoli in control group was regular on day 7 with even sizes (Fig. 1A). On day 14, the structure of pulmonary alveoli was normal, and there was narrowing in the alveolar septum in control (Fig. 1B). On day 21, there was increase in number of pulmonary alveoli, together with narrowing of alveolar septum. There were no aberrant changes in terminal bronchus in control group (Fig. 1C). For the H&E staining in BPD model group, part of pulmonary alveoli showed fusion on day 7, combined with infiltration of inflammatory cells (Fig. 1A). On day 14, the number of pulmonary alveoli showed decrease and the structure was not regular. There was massive interstitial cell hyperplasia, together with thickening in alveolar septum (Fig. 1B). On day 21, pulmonary alveoli were no longer available, and there was obvious dilatation in terminal bronchus. In addition, structural disorder was noticed in pulmonary tissues, indicating block in pulmonary development (Fig. 1C).

Figure 1 H&E staining of lung tissue under a magnification of ×100.

(A–C) H&E staining of lung tissue on day 7, 14 and 21. BPD: bronchopulmonary dysplasia.

Alpha diversity

Species accumulation curve was used to confirm whether the sample was sufficient. In the species accumulation curve, the growth rate of magenta box line reflected the emergence rate of new species under continuous sampling. When the curve was tended to be flat, it meant that there were enough samples to reflect the number of species in the environment. The green box curve represented the number of species common to each sample. With the increase of sampling samples, the green box curve decreased. When the curve was tended to be flat, it meant that the same species in the environment also tend to be stable. According to our species accumulation curve, the magenta and green curves have tended to be flat, indicating that the number of samples is sufficient for data analysis (Fig. 2A). Multi-sample Shannon curves indicated that the data volume was adequate for the sequencing, and the sample traits would not increase with the elevation of sequencing volume (Fig. 2B). Alpha diversity was analyzed to evaluated overall differences between the gut microbiota in model group and control group. The ACE and Chao1 index showed no statistical differences in richness of gut microbiota between control group and model group on day 7, 14, and 21, respectively (p > 0.05, Figs. 2C and 2D). This implied that there were no statistical differences in the richness of gut microbiota between two groups.

Figure 2 Alpha diversity reflected richness and diversity of bacterial communities.

(A) Species accumulation curve of the sample at the genus level. The curve was tended to be flat, indicating sufficient sampling. (B) Shannon index curve of samples at different time. Curve was flat, and amount of sequencing tended to be saturated, which could reflect the biological diversity of the samples. (C and D) Student’ s t-test was used to test significance based on ACE and Chao1 index. BPD, bronchopulmonary dysplasia.

Beta diversity

In this section, Beta diversity analysis was performed based on un-weighted unifrac distance. PCoA plot showed that there was no obvious separation between two groups on day 7 and day 21, respectively. In contrast, there was significant separation of PC1 between two groups on day 14 (Figs. 3A–3C). Analysis of similarities indicated that there was no significant difference in gut microbiota between two groups on day 7 (R = −0.028, p = 0.628), while the difference was statistically significant between two groups on day 14 (R = 0.368, p = 0.021). On day 21, there was no difference in gut microbiota between two groups (R = 0.188, p = 0.079, Table 2).

Figure 3 PCoA analysis based on un-weight Unifrac distance between groups.

(A–C) PCoA analysis on day 7, 14 and 21. PCoA, principal coordinates analysis; BPD, bronchopulmonary dysplasia. The red dots represent Control and the green dots represent BPD.

Table 2 Analysis of similarities.

Variable	Day 7	Day 14	Day 21	
Sample similarity within groups				
All	0.212 ± 0.049	0.298 ± 0.035	0.236 ± 0.043	
Control	0.243 ± 0.044	0.348 ± 0.065	0.265 ± 0.034	
BPD	0.181 ± 0.031	0.248 ± 0.038	0.207 ± 0.031	
Sample similarity between groups	0.204 ± 0.042	0.341 ± 0.035	0.246 ± 0.028	
R value	−0.028	0.368	0.188	
p value	0.628	0.021	0.079	
Note:

BPD, bronchopulmonary dysplasia.

Difference analysis

LEfSe data analysis was performed to investigate the biomarkers, and all biomarkers had linear discriminant analysis (LDA) scores of higher than 4. There were three biomarkers with statistical differences at all biological levels on day 7, all of which were in the control group (p < 0.05). On day 14, there were statistical differences in 22 biomarkers at each biological level, among which 16 were enriched in BPD group and six were enriched in control group (p < 0.05, Figs. 4A and 4B). On day 21, there were statistical differences in four biomarkers at all biological levels, all of which were in BPD group (Fig. 4C).

Figure 4 LDA effect size showed the different microbiota from the kingdom level to the species level between groups (LDA score >4 and P < 0.05).

(A) Three biomarkers were significantly different in the control group on day 7; (B) Twenty-two biomarkers were significantly different between groups on day 14; (C) Four biomarkers were significantly different in the BPD model group on day 21. LDA, linear discriminant analysis; BPD, bronchopulmonary dysplasia.

On day 14, the relative abundance of intestinal microbiota showed that the proportion of Firmicutes, Bacteroidetes and Proteobacteria in phylum level was higher than 80% (Fig. 5). Analysis of metastats on phylum level indicated that the relative richness of Bacteroidetes in model group was significantly lower than that of control group (11.6% vs 54.8%, PFDR < 0.01), while the relative richness of Proteobacteria in model group was significantly higher than that of control group (29.8% vs 5.1%, PFDR < 0.05). This was consistent with the LEfSe results. The relative richness of Cyanobacteria, Acidobacteria, Chloroflexi, Rokubacteria, Epsilonbacteraeot, Nitrospirae and Gemmatimonadetes in model group was significantly higher than that of control (all PFDR < 0.05). The analysis of metastats on phylum level was shown in Table S2.

Figure 5 The relative abundance of intestinal microbiota at the phylum level in different periods.

Only 10 groups of microbiota with the highest relative content were shown in the picture, and other intestinal microbiota were classified as Others. BPD, bronchopulmonary dysplasia.

Functional prediction

A total of 17 KEGG metabolic pathways associated with intestinal microbiota were significantly different between the two groups. Three of them were enriched in the control group and 14 were enriched in the model group. Signal transduction (PFDR = 0.037), glycan biosynthesis and metabolism (PFDR = 0.032), metabolism of terpenoids and polyketides (PFDR = 0.049) are the top three metabolic pathways with statistically significant differences (Fig. 6).

Figure 6 KEGG pathway analysis of the intestinal microbiota.

Three pathways were enriched in control model group, and fourteen pathways were enriched in BPD group. BPD: bronchopulmonary dysplasia.

Discussion

Patients with intestinal diseases may present symptoms in respiratory system, while those with respiratory diseases may accompany by intestinal symptoms (Neurath, 2020; Ojha et al., 2018; Wang et al., 2014). Thus, there is a close interaction between respiratory and intestinal diseases (Crawford, Nordgren & McCole, 2021; Raftery et al., 2020), which is defined as gut-lung axis. To our best knowledge, the microorganism is crucial for the development of immune system and metabolic balance in hosts (Budden et al., 2017; Sarkar et al., 2021; Spielman, Gibson & Klegeris, 2018). The microflora in gut and lung play important roles in the pathogenesis of pulmonary and intestinal diseases through the gut-lung axis (Chioma et al., 2021; Deriu et al., 2016; Wypych, Wickramasinghe & Marsland, 2019). However, there is a lack of studies on changes of gut microbiota in neonates with BPD.

Our study was designed to investigate the changes of gut microbiota in BPD model. We established the BPD model of neonatal mice by hyperoxia environment, to detecting the changes of gut microbiota and the damage degree of lung tissue at different time. Thus, we can clearly demonstrate the relationship between the intestinal microbiota and the BPD model. On day 7, the anti-oxidant system in the pulmonary tissues was immature, and the anti-infection and immune system development were not well developed. Studies have shown that there would be blockage in alveolarization in lung-term exposure of high concentration of oxygen (Yu et al., 2020). Additionally, there was massive generation of TNF-α, IL-6, IL-8 and MCP-1 in lung tissues (Bhandari, 2010) and inflammatory reactions in lung tissues. Our results showed that the lung tissue in model group has been destroyed compared with the control group on day 7. However, there was no significant difference in intestinal microbiota between two groups on day 7. This indicates that the destruction of lung tissue occurs before the change of intestinal microbiota in the BPD mouse model. On day 14, H&E staining showed serious injuries in lung tissues than before, and the differences in gut microbiota between two groups were statistically significant (P < 0.05). While the relative richness of Bacteroidetes was decreased in gut microbiota of model group, the relative richness of Proteobacteria was increased. KEGG pathway analysis indicated that there were statistical differences in signal transduction, glycan biosynthesis and metabolism, as well as metabolism of terpenoids and polyketides between two groups. This indicated that BPD rats showed disorder in the gut microbiota associated with regulation of signal transduction and metabolism. Nevertheless, the exact mechanisms are still unclear. The hyperoxic environment would induce damages to alveolar septum (Dauger et al., 2003) and there might be persistent inflammatory response in lung tissues (Balany & Bhandari, 2015), which further aggravated lung injury. On day 21, the pulmonary alveoli was no longer available in the pulmonary tissues of BPD model, and the structure in pulmonary tissues was not regular. In control group, the development of pulmonary tissues was normal. However, there was no statistical difference in gut microbiota between two groups. Here are some reasons, the composition of gut microbiota was affected by diet, age, development, genetics, and antibiotics (Jacobs et al., 2020; Lynch & Pedersen, 2016). In cases of any changes of diet, there would be rapid spontaneous remodeling for gut microbiota (Kau et al., 2011). In our study, mice were randomly divided into different groups after birth, without exposure to any antibiotics. All the animals were free access to a diet and water on day 12–21, followed by termination of lactation. On this basis, we hypothesized that with the increase of age, there would be gradual maturity for gut development on day 21. In a previous study, there would be a gut microbiota balance in mice since termination of lactation (Pantoja-Feliciano et al., 2013). After spontaneous intake of diet, the uptake of fiber showed increase, which promoted the stability of gut microbiota (Conte & Toraldo, 2020) and reduction of differences in gut microbiota between two groups.

In our experiment, lung tissue damage in BPD mice occurred before gut microbiota alterations, and the gut microbiota of the two groups appeared significantly different on day 14, which may be related to the regulation of signal transduction and metabolism. It has been noted that intestinal microbes can have effects on lung tissue, disruption of the gut microbiota promotes a more severe BPD phenotype (Willis et al., 2020). This situation where the gut and lung interact with each other is called the gut-lung axis. In newborn mice, there is a critical developmental period during which the gut microbiota can guide the pulmonary transport of innate lymphocytes (ILC3) and affect the susceptibility to bacterial pneumonia (Gray et al., 2017). Microorganisms could regulate immune reactions in intestinal and pulmonary tissues through modulating NLRP3 inflammatory bodies, which then affect intestinal and pulmonary disorders (Donovan et al., 2020). Short chain fatty acid generated by gut microbiota would regulate immune balance in lung tissues (Depner et al., 2020). Breast milk may involve in prevention of BPD through affecting formation of microorganisms and regulating inflammatory reactions (Piersigilli et al., 2020). The mechanism of intestinal microbes affecting lung tissue is very complex, and in the future, more studies are required to illustrate the potential mechanism. Changes of microorganism at early stage would affect the lung response in male mice responding to environmental changes (Brown et al., 2020). In our study, the gut microbiota of the two groups of mice was significantly altered on day 14, and the difference gradually disappeared on day 21. We speculate that there is a certain possibility to delay or alleviate the progression of BPD by regulating gut microbiota at the early stage of BPD. Therefore, further studies are required to further investigate whether early-stage interference to gut microbiota would affect the progress of BPD.

There are some limitations in this study. There was no grouping based on gender in this study. Although the changes of pulmonary tissues in BPD mice and control mice were similar, there might be differences in microorganism formation and microbiota between neonatal mice and neonates (Hildebrand et al., 2013). In addition, considering the differences in development time between humans and mice, it is necessary to verify these data in humans. Therefore, verification is our next effort in clinical practice.

Conclusions

Our data proved the change of lung tissues before the change of intestinal microbiota in the model group. There was alteration of gut microbiota in BPD mice on day 14. Specifically, the proportion of Bacteroidetes and Proteobacteria showed significant changes, which may be related to the signal transduction and metabolic signaling pathways in the early stage. With the progression of neonatal maturity, the gut microbiota gradually stabilized. Therefore, whether early changes in the gut microbiota can reduce or delay the progression of BPD requires further research.

Supplemental Information

Supplemental Information 1 The statistics in the process of processing reads.

Click here for additional data file.

Supplemental Information 2 Metastats on phylum level between two groups on day 14.

Click here for additional data file.

Supplemental Information 3 ARRIVE 2.0 Checklist.

Click here for additional data file.

Abbreviations

BPD bronchopulmonary dysplasia

H&E hematoxylin and eosin

LDA linear discriminant analysis

LEfSe linear discriminant analysis effect size

SPF specific-pathogen free

Anosim analysis of similarities

Additional Information and Declarations

Competing Interests

Author Contributions

Animal Ethics

DNA Deposition

Data Availability

The authors declare that they have no competing interests.

Tianqun Fan analyzed the data, authored or reviewed drafts of the paper, and approved the final draft.

Ling Lu conceived and designed the experiments, authored or reviewed drafts of the paper, and approved the final draft.

Rong Jin performed the experiments, prepared figures and/or tables, and approved the final draft.

Aihua Sui analyzed the data, prepared figures and/or tables, and approved the final draft.

Renzheng Guan analyzed the data, prepared figures and/or tables, and approved the final draft.

Fengjing Cui analyzed the data, prepared figures and/or tables, and approved the final draft.

Zhenghai Qu conceived and designed the experiments, authored or reviewed drafts of the paper, and approved the final draft.

Dongyun Liu conceived and designed the experiments, authored or reviewed drafts of the paper, and approved the final draft.

The following information was supplied relating to ethical approvals (i.e., approving body and any reference numbers):

The study protocols were approved by the Ethical Committee of Affiliated Hospital of Qingdao University (Approval No.: QYFYWZLL26150)

The following information was supplied regarding the deposition of DNA sequences:

The metagenome sequences of the gut microbiota of mice are available at SRA: PRJNA743668.

The following information was supplied regarding data availability:

The data is available at Zenodo: Tianqun Fan. (2021). Dataset of intestinal microbiota in mice model of bronchopulmonary dysplasia [Data set]. Zenodo. https://doi.org/10.5281/zenodo.5031783.

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
