# Peer review of "Change of intestinal microbiota in mice model of bronchopulmonary dysplasia"

_PeerJ, doi:10.7717/peerj.13295_

## Round 0.1 · original submission · Major Revisions

Please, revise carefully the comments provided by the reviewers. The revision does not guarantee the potential acceptance of this work.

·

Basic reporting

No comments

Experimental design

1. Write HE staining as hematoxylin and eosin (H&E), if it appears for the first time in the manuscript.
2. For the induction of BPD neonatal mice were subject to high concentration of oxygen (80±5%) for 3 weeks. How did the authors reach to these figures. What about other concentrations?
3. Mention the mice species in the M & M
4. The authors are suggested to submit the data to appropriate data bank and mention the accession number.
5. Validation with human subject is suggested.

Validity of the findings

1. How can we reach to the species level with a single taxonomic marker i.e., 16S ? Why microbiota study upto species level was not performed?
2. There is no mention of fungal diversity.
3. Justify, how the experiments done in animal model can be used to validate the diversity of microbial diversity in human infants.

Reviewer 2 ·

Basic reporting

1. The disease in question is bronchopulmonary dysplasia. There is no hyphen.
2. Its Linear discriminant analysis not line discriminant analysis.
3. The paper completely fails to discuss the relevant literature. Since all of these analysis have previously been published in both humans and mice, basic scientific practice would be to discuss this in the introduction and discussion, as this is the point of those sections of a scientific paper. Some examples by pubmed ID: PMID: 31644311, PMID: 32877224, PMID: 31644312, PMID: 27488092, PMID: 32801143, PMID: 31664856, PMID: 32117822, PMID: 30487069, PMID: 31967848, PMID: 31898918.
4. The microbiome data in the tables is of limited value and should likely be placed in the supplement.

Experimental design

There are significant limitations in reporting of rigorous research and in study design.
1. Since this study replicates a portion of this study in AJP Lung (PMID: 31644311), I would suggest starting there to see the level of detail needed in reporting the model design and microbiome analyses.
2. The most significant issue is in statistical power. In neonatal microbiome studies the biological replicate is the Litter, not the individual mouse as all pups in the same litter are effectively technical replicates. Therefore the effective samples size is 1-2 for most of these experiments. The whole paper needs to be repeated with markedly more samples. PMID: 32688019
3. In figure 1, the representative histology does not appear to have been prepared correctly as the samples appear under inflated, this calls into question whether the authors actually performed the hyperoxia exposure as described. Since this model is well published norms are available to quantify these values. I suggest following the procedures in the ATS guidelines (PMID: 23624789) and the norms in the paper by the Morty lab: PMID: 29315540.

Validity of the findings

1.Validity is uncertain as the study is vastly underpowered - see point 2 in Experimental Design
2. Sequencing data must be uploaded to a public repository.

---

## Round 0.2 · Major Revisions

The manuscript has been checked and still major comments need to be covered. Please, revise carefully all comments raised by reviewers.
This second revision does not guarantee the publication of your work.

Reviewer 3 ·

Basic reporting

This is a nice experimental paper investigating the importance of the gut-lung axis microbiota in the development of BPD. The paper is well designed, nonetheless there are important issues, mainly in the statistical power, that make the paper unsuitable to be published in the present form.



English is overall acceptable, but sometimes the grammar and construction of the phrases makes it hard for the reader ti follow the reasoning or the meaning of the sentence. For example:
"Compared with the control group, the development of lung tissue in model group has been destroyed. " what does this sentence mean?
"The microflora in gut and lung are rather complicated, which play important roles in the pathogenesis
of pulmonary and intestinal diseases through the gut-lung axis": the sentence is not clear, please rephrase
Line 255: "alternation" Do the authors mean "alteration"?
Line 258: "In the presence of neonatal maturity, the gut microbiota gradually stabilized." the authors mean " with the progression of neonatal maturity..."?

Therefore the paper has to be revised by a native speaker to improve the quality of the writing and the reasoning

Literature references are overall sufficient, but as the authors are referring to a BPD model , the "new BPD" has to be cited. In fact the authors refer to the development of BPD related to ventilation and oxygen toxicity, but they have to cite also the changes in vascular structure !that are now the main cause of BPD in extremely preterm neonates. (Bronchopulmonary Dysplasia: 50 Years after the Original Description. Bancalari E, Jain D. Neonatology. 2019;115(4):384-391. doi: 10.1159/000497422. Epub 2019 Apr 11. PMID: 30974430)

Raw data are correct tell shared

Experimental design

The most significant issue is the statistical power. the experiments have been carried out with a limited amount off samples (5 for each group) Therefore the results have not enough statistical power to be accepted for publication. The experiments should be repeated with more samples as to confirm the results obtained so far. The paper could be accepted as preliminary data foreseeing a more extensive analysis.

Ethical standards were correctly met.

Methods were correctly described.

Validity of the findings

As commented previously, the main issue is the insufficient sample size.

Sometimes data are confusing and contradictory: for example line 195: "there was no significant difference in intestinal microbiota between two groups" opposed to line 200 "and the differences in gut microbiota between two groups were statistically significant (P<0.05)."

Is is not clear if some of the findings discussed in the paper were analyzed in the paper or are just referred from other paper (for example "There was massive generation of TNF-α, IL-6, IL-8 and MCP-1 in lung tissues(Bhandari 2010) " was this confirm by the paper or just cited?)

The authors found 3 different metabolic pathways: Signal transduction, glycan biosynthesis and metabolism , metabolism of terpenoids and polyketides. The authors should extensively comment on the importance of these pathways in the discussion.

Additional comments

With the proposal of the gut-lung axis concept(Budden et al. 2017).: the sentence has to be completed

Line 237 "As previously described, microorganism would affect the progress of certain diseases rather than improving their severity" what do the authors exactly mean? Please explain better

Conclusion:
the is no need to repeat the experimental model her "A BPD model was established in neonatal mice through exposure of hyperoxic environment. Then we analyzed the difference of gut microbiota diversity between natural mice and BPD model mice"

The authors should comment on the potential importance of probiotics to change the intestinal microbiota and reduce the incidence of BPD.

---

## Round 0.3 · accepted · Accept

Authors have covered the reviewer´s questions and concerns.

Reviewer 3 ·

Basic reporting

the reviewers addressed my concerns in the new version of the paper

Experimental design

the experimental design is clear

Validity of the findings

my concerns on number of the samples have been cleared

Additional comments

the reviewers addressed my concerns in the new version of the paper